# Individuals with ventromedial frontal damage display unstable but transitive preferences during decision making

**Linda Q. Yu** [1,2] ✉, **Jason Dana**[3] **& Joseph W. Kable** [1]

The ventromedial frontal lobe (VMF) is important for decision-making, but the precise causal role of the VMF in the decision process has not been fully established. Previous studies have suggested that individuals with VMF damage violate transitivity, a hallmark axiom of rational decisions. However, these prior studies cannot properly distinguish whether individuals with VMF damage are truly prone to choosing irrationally from whether their preferences are simply more variable. We had individuals with focal VMF damage, individuals with other frontal damage, and healthy controls make repeated choices across three categories—artworks, chocolate bar brands, and gambles. Using proper tests of transitivity, we find that, in our study, individuals with VMF damage make rational decisions consistent with transitive preferences, even though they exhibit greater variability in their preferences. That is, the VMF is necessary for having strong and reliable preferences, but not for being a rational decision maker. VMF damage affects the variability with which value is assessed, but not the consistency with which value is sought.

A central assumption of rational choice theories is that decision-makers compare the subjective value of different options and choose the highest valued option. Satisfying this assumption is equivalent to having transitive preferences[1]. An example of transitivity is the following: If you prefer to listen to Adele (A) over Beyoncé (B), and Beyoncé over Celine Dion (C), then you would also prefer Adele (A) over Celine (C). There is a strong argument that preferences ought to be transitive, as an intransitive chooser would get caught in choice cycles that do not advance towards any goal (choosing A over B, B over C, and C over A) and that could be exploited (e.g., an unsavory ticket hawker can keep charging you to trade for tickets to your ever-shifting more preferred artist). Thus, nearly all normative theories of decision-making are transitive.

Given this, one might expect that organisms have internal representations of value that are transitive. Dozens of functional neuroimaging studies in humans and neurophysiological studies in non-human animals have now identified neural activity in the ventromedial frontal lobe (VMF) that scales with subjective value across different

categories of goods[2–5]. Correspondingly, lesions to the VMF impair value-based decision-making in a variety of ways[6–10].

However, the precise role of the VMF in rational choice is still unclear. Intriguingly, several previous studies have shown that individuals with VMF damage make more cyclical choices (i.e., choosing C over A after previously selecting A over B and B over C) than healthy controls or individuals with damage elsewhere in the frontal lobe[11–13]. This increase in cyclical choices after VMF damage, however, is consistent with two very different possibilities regarding the necessary role of VMF, and putative value signals in VMF, in rational choice.

The first possibility is that the preferences of individuals with VMF damage are fundamentally intransitive and not self-consistent. In this case, in the example above, the individual with VMF damage would consistently and reliably choose C over A. This could occur if individuals with VMF damage choose according to stimulus-response associations or rules that lack any higher order transitive structure, rather than according to any set of underlying preferences. If this possibility were true, the proper conclusion would be that an intact VMF is

[1]Department of Psychology, University of Pennsylvania, Philadelphia, PA 19104, USA. [2]Department of Neuroscience, Brown University, Providence, RI 02912, USA. [3]Yale School of Management, Yale University, New Haven, CT 06520, USA. ✉e-mail: linda_yu@brown.edu

necessary for the human brain to assess value at all, or to use value to make decisions; i.e., that an intact VMF is necessary for transitive preferences and rational choice.

The second possibility is that the preferences of individuals with VMF damage are transitive and self-consistent, but more variable. In this case, the individual in the example above might prefer A over C on average, but less decisively than others. Thus, if asked again later, this individual would have a greater chance of changing their mind and now choosing C over A. This change of mind could occur if individuals with VMF damage choose according to underlying values, but the assessment of those values was more variable across time. If this possibility were true, the proper conclusion would be that an intact VMF promotes the stability and reduces the variability of valuations across time; but an intact VMF is not necessary for transitive preferences and rational choice.

Distinguishing between preferences that are fundamentally intransitive and preferences that are simply more variable is a deep problem in testing theories of rational choice that has only recently been solved for the example of transitivity by Regenwetter and colleagues[14]. Importantly, these authors recognize that behavior in experiments is probabilistic, and therefore testing axioms of rational choice like transitivity requires recasting these axioms in probabilistic terms[14–17]. Critically, counting the number of observed choice cycles (e.g., choosing C over A when one has chosen A over B and B over C), as done in previous studies, is unrelated to the degree to which choices violate a probabilistic model of transitivity. Put differently, counting choice cycles does not help to disentangle whether one has fundamentally intransitive versus variable preferences. We use these recently developed tests of a probabilistic model of transitivity to determine whether the preferences of individuals with VMF damage are more variable or fundamentally intransitive. If the choices of individuals with VMF damage do not satisfy these tests, we would have evidence that their preferences are fundamentally intransitive. Alternatively, if their choices do satisfy such tests, it would suggest that previous findings of more choice cycles are due to variability in preference (a possibility that can be further confirmed with additional modeling). A definitive discrimination between these two possibilities will identify more precisely the necessary role of VMF, and putative value signals in VMF, in value-based decision-making.

## Results

We tested thirteen individuals with VMF damage, and for comparison, ten individuals with damage to frontal lobe (including dorsomedial, dorsolateral, and insular areas) other than VMF (referred to as frontal controls, or FC), and twenty age and education-matched healthy controls (HC; see Methods for inclusion criteria). Figure 1 shows the overlap of lesions in the two lesion groups, and Table 1 provides the demographic characteristics of all three groups.

Participants made binary preference decisions in three categories: artworks, chocolate bar brands, and monetary gambles. Each category had two sets of options. Set A, used for probabilistic tests of transitivity, contained 5 items that were used to construct 10 binary choices that were each repeated 15 times throughout the experiment. Set B, used to test for choice cycles, contained 10 or 11 items (10 for chocolate bar brands, 11 for artworks and gambles) that were used to construct 45 or 55 binary choices that were each presented once. Choices using set A and set B items were intermingled in each block. All items in the artwork and chocolate bar brand sets were normed to be as close in value as possible, and all gambles were of equal expected value (see "Methods" for details).

### Most individuals, and all individuals with VMF damage, make choices consistent with a probabilistic model of transitivity

Our central question is whether the choices of individuals with VMF damage would violate a probabilistic model of transitivity. If they do, this would show that the preferences of individuals with VMF damage are fundamentally intransitive; if they do not, this would suggest that their preferences might be more variable, but are not intransitive.

To answer this question, we examined the subset of choices in our experiment, set A, which involves 15 repetitions each of 10 different binary choices in each of the three categories. In each category, we can calculate 10 choice percentages, one for each of the possible pairings of five items. Table 2 shows these choice percentages in each category for four example participants with VMF damage. Here the items are ranked according to the number of times they are chosen overall, and the choice percentages are coded such that a percentage greater than 50% indicates more choices of the higher ranked item. Two things are notable about these choice percentages. First, most percentages are less than 100%, demonstrating the need for a probabilistic model of choice. Second, most percentages are greater than 50%, suggesting that a probabilistic model of transitive choice may be appropriate for these data.

These data are sufficient to evaluate whether the choices each participant made are consistent with a probabilistic model of transitive choice called the mixture model. The mixture model[14,18,19] assumes that every choice is made according to a preference ordering, but that the preference ordering governing a specific choice is drawn randomly from a mixture of all possible preference orderings. Regenwetter and colleagues[14] show that these assumptions place a constraint on the observable choice percentages called the triangle inequalities (see "Methods") and develop novel statistical methods for determining when violations of the triangle inequalities are unlikely to be due to random sampling. For five stimuli and thus ten pairwise choice

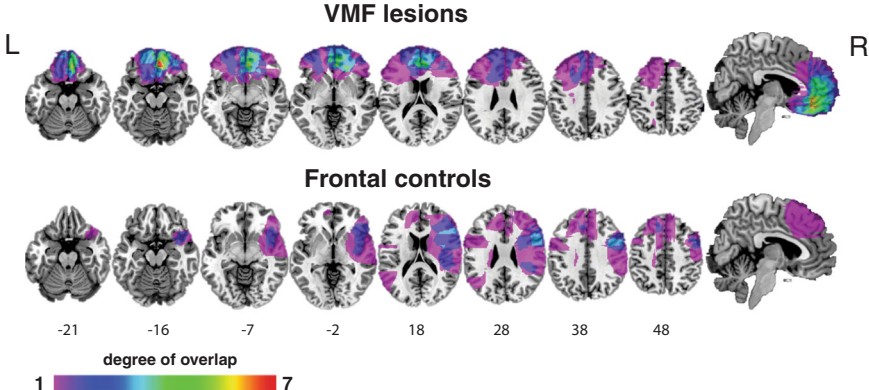

**Fig. 1 | Overlap of lesions in the VMF and frontal control groups.** Numbers below slices indicate the MNI z-coordinates. Colors indicate extent of overlap. L = left; R = right.

percentages, the triangle inequalities fully characterize the mixture model and impose rather restrictive constraints; only 5% of the sample space satisfies the triangle inequalities.

Across the individuals tested, most choices—including all of those from individuals with VMF damage—were consistent with this probabilistic model of transitivity (124 of 129 total tests across all individuals and domains, Table 3). None of the individuals with VMF damage significantly violated the mixture model in any of the three domains (out of a total of 39 tests, Table 3). One of the individuals in the FC group significantly violated the mixture model in the gambles domain (out of a total of 30 tests, Table 3). Four individuals in the HC group significantly violated the mixture model in the gambles domain, and one in the brands domain (out of a total of 60 tests, Table 3).

**Individuals with VMF damage have more variable preferences**
We find that the choices of individuals with VMF damage are consistent with a probabilistic model of transitivity. This suggests that the previously documented tendencies of these individuals may be due to having preferences that are more variable, rather than preferences that are intransitive. That is, their choices reflect underlying transitive preference orderings, but they vacillate among preference orderings

more than other choosers. However, an alternative explanation is that the individuals that we tested with VMF damage do not exhibit the same alterations in decision-making that have been documented previously in other individuals with VMF damage. To rule out this alternative explanation, we performed two additional analyses to show that the preferences of individuals with VMF damage in our study are more variable and not unlike those from previous studies.

First, to directly estimate preference variability, we fit each individual's choices and response times (RTs) in each category to a drift diffusion model (DDM)[20]. We fit the DDM to the same choices used to test the mixture model. The DDM we fit assumes that choices and RTs are a probabilistic function of the rank distance between the two options in the average preference ordering for that individual (see "Methods" for details on how the average preference ordering was determined). Specifically, we modelled the decision process as resulting from a decision variable (DV), which starts at an initial value (*int*) that is constant across trials, and then, after a period of non-decision time (ndt), increases linearly across time with a slope $d*v^\alpha$, where $d$ is the drift rate, $v$ is the rank difference between the two items, and $\alpha$ is an exponent accounting for potential non-linearities in the effect of rank difference. At each time step, there is also Gaussian variability with a standard deviation of $\varepsilon$ added to the DV, which is the key parameter of interest in our case (see Fig. 2A for a graphical illustration of this model). A decision is made when the DV crosses a fixed threshold value corresponding to a choice of the higher or lower ranked item.

Fits to this model revealed that individuals with VMF damage were more variable choosers. We conducted a 2-way mixed measures ANOVA for each parameter of the DDM, with item category (artworks,

### Table 1 | Demographics of participants

| Group (n) | Gender | Mean age (sd) | Education in yrs (sd) |
|---|---|---|---|
| VMF (13) | 7F:6M | 59 (15) | 14 (3) |
| FC (10) | 7F:3M | 66 (8) | 14 (3) |
| HC (20) | 15F:5M | 62 (8) | 15 (2) |

### Table 2 | Choice percentages (in set A) for four individuals with VMF damage

| | Category | | | | | | | | | | | | | | | |
|---|---|---|---|---|---|---|---|---|---|---|---|---|---|---|---|---|
| | Art | | | | | Brands | | | | | Gambles | | | | | |
| **VMF2** | A | B | C | D | E | A | B | C | D | E | A | B | C | D | E |
| A | | | | | | A | | | | | A | | | | | |
| B | 93 | | | | | B | 67 | | | | B | 100 | | | | |
| C | 100 | 67 | | | | C | 60 | 40 | | | C | 87 | 93 | | | |
| D | 100 | 80 | 33 | | | D | 73 | 87 | 60 | | D | 100 | 100 | 73 | | |
| E | 93 | 67 | 80 | 47 | | E | 80 | 87 | 73 | 67 | E | 93 | 100 | 100 | 87 | |
| **VMF8** | A | B | C | D | E | A | B | C | D | E | A | B | C | D | E |
| A | | | | | | A | | | | | A | | | | | |
| B | 67 | | | | | B | 60 | | | | B | 33 | | | | |
| C | 80 | 40 | | | | C | 93 | 80 | | | C | 67 | 47 | | | |
| D | 100 | 100 | 80 | | | D | 73 | 67 | 80 | | D | 67 | 67 | 47 | | |
| E | 100 | 93 | 100 | 67 | | E | 80 | 93 | 80 | 40 | E | 80 | 53 | 80 | 67 | |
| **VMF10** | A | B | C | D | E | A | B | C | D | E | A | B | C | D | E |
| A | | | | | | A | | | | | A | | | | | |
| B | 60 | | | | | B | 93 | | | | B | 93 | | | | |
| C | 53 | 67 | | | | C | 67 | 67 | | | C | 93 | 73 | | | |
| D | 60 | 60 | 60 | | | D | 100 | 100 | 40 | | D | 80 | 67 | 73 | | |
| E | 100 | 100 | 100 | 93 | | E | 100 | 100 | 100 | 60 | E | 100 | 80 | 100 | 93 | |
| **VMF13** | A | B | C | D | E | A | B | C | D | E | A | B | C | D | E |
| A | | | | | | A | | | | | A | | | | | |
| B | 60 | | | | | B | 60 | | | | B | 40 | | | | |
| C | 53 | 53 | | | | C | 60 | 47 | | | C | 87 | 73 | | | |
| D | 67 | 73 | 67 | | | D | 60 | 80 | 47 | | D | 73 | 67 | 73 | | |
| E | 60 | 67 | 73 | 53 | | E | 67 | 53 | 67 | 60 | E | 93 | 67 | 73 | 60 | |

The complete set of choice percentages in each stimulus category in set A, for each of four individuals with VMF damage. These are the same four individuals who had significantly more choices cycles than healthy controls in Fig. 3. The options are ranked A-E, where A is the option that was chosen most often by that subject, B is the option chosen second most often, etc., and the numbers are the percentage of choices where the column option is chosen over the row option.

## Table 3 | Results of LOP analysis by category

| Respondent | Artworks p-value | Brands p-value | Gambles p-value |
|---|---|---|---|
| **Individuals with VMF damage** | | | |
| 1 | √ | √ | √ |
| 2 | 0.68 | √ | 0.05 |
| 3 | √ | √ | 0.33 |
| 4 | √ | √ | √ |
| 5 | √ | √ | √ |
| 6 | √ | √ | √ |
| 7 | √ | 0.33 | √ |
| 8 | √ | √ | √ |
| 9 | √ | √ | 0.42 |
| 10 | √ | √ | √ |
| 11 | √ | √ | √ |
| 12 | √ | √ | √ |
| 13 | √ | √ | √ |
| **Frontal controls** | | | |
| 1 | 0.09 | √ | √ |
| 2 | √ | √ | √ |
| 3 | √ | √ | √ |
| 4 | √ | √ | √ |
| 5 | √ | √ | √ |
| 6 | √ | √ | 0.33 |
| 7 | √ | √ | **0.00** |
| 8 | √ | √ | 0.21 |
| 9 | √ | √ | 0.11 |
| 10 | √ | √ | √ |
| **Healthy controls** | | | |
| 1 | √ | 0.54 | **0.00** |
| 2 | √ | √ | 0.44 |
| 3 | √ | √ | √ |
| 4 | √ | √ | √ |
| 5 | √ | √ | √ |
| 6 | √ | 0.83 | √ |
| 7 | √ | √ | √ |
| 8 | √ | √ | 0.42 |
| 9 | √ | √ | **0.01** |
| 10 | √ | √ | **0.00** |
| 11 | √ | √ | √ |
| 12 | √ | √ | 0.07 |
| 13 | √ | √ | √ |
| 14 | √ | √ | √ |
| 15 | 0.07 | √ | √ |
| 16 | √ | **0.02** | **0.00** |
| 17 | √ | √ | √ |
| 18 | √ | 0.14 | √ |
| 19 | √ | 0.10 | 0.19 |
| 20 | √ | √ | √ |

Note: Each participant made choices in all three categories.
Checkmark indicates subject fulfilled triangle inequalities (*p* = 1) for that category. *p*-values reflect results of a goodness-of-fit test for the linear ordering polytope using a chi-bar-square distribution and are not adjusted for multiple comparisons; significant violations are marked in bold.

brands, gambles) as a within-subject factor and group (VMF, FC, or HC) as a between-subjects factor. The only parameter of the DDM that was significantly different across groups was the variability parameter $\varepsilon$ [$F(2,37) = 6.25, p = 0.005$]. Specifically, the VMF group (mean = 0.12, sd = 0.03) had significantly higher $\varepsilon$ than HC (mean = 0.09, sd = 0.04)

[$t(28) = 2.08, p = 0.047$] and FC (mean = 0.07, sd = 0.02) [$t(20) = 3.94$, p < 0.001]. No other parameters differed between the three groups [drift rate, *d*: $F(2,37) = 1.78, p = 0.18$; non-linearity, α: $F(2,37) = 0.45$, $p = 0.64$; starting point, *int*: $F(2,37) = 0.24, p = 0.79$; non-decision time, ndt: $F(2,37) = 0.07, p = 0.94$; Fig. 2B]. The complete set of results from the ANOVAs on the parameters are reported in the Supplementary Materials.

A model-free examination of RTs further supports the conclusion that individuals with VMF damage differ from healthy controls specifically in the variability parameter in the DDM (Fig. 2C). An increase in the variability parameter in the DDM causes faster RTs both for choices consistent with the average preference ranking (i.e., "correct" choices) and for choices inconsistent with the average preference ranking (i.e., "errors"). In contrast, an increase in the drift rate in the DDM only speeds "correct" choices. Indicative of an increase in variability, individuals with VMF damage exhibited numerically, though not significantly, faster RTs for "correct" choices (VMF mean correct RT = 1.86 s, sd = 0.43; HC mean correct RT = 2.32 s, sd = 0.90; $t(28)=1.3$, $p = 0.35$) and significantly faster RTs for "error" choices compared to HCs (VMF mean error RT = 4.82 s, sd = 0.91; HC mean error RT = 6.37 s, sd = 2.10; $t(28) = 2.29, p = 0.03$).

### Individuals with frontal damage exhibit more choice cycles

In addition to showing that preferences are more variable in individuals with VMF damage, we also sought to replicate the previously documented tendency of individuals with VMF damage to exhibit more choice cycles (11, 12). Although we do not endorse counting cycles as a measure of transitive choice, given the problems identified with this measure[14,21], such replication would provide evidence that our individuals with VMF damage are not behaving differently than individuals with VMF damage in prior studies. We examined the second subset of the choices in our experiment, set B, which consisted of a single instance of all pairwise choices between a total of ten or eleven items in each category.

We replicated the finding that individuals with VMF damage have more choice cycles than HC individuals (Fig. 3). We first conducted a 2-way mixed measures ANOVA of the number of choice cycles made by each participant, with item category (art, brands, gambles) as a within-subject factor and group (VMF, FC, or HC) as a between-subjects factor. We found a significant main effect of group [$F(1, 41) = 6.28$, $p = 0.016$], but no main effect of category ($p = 0.71$) nor a group x category interaction ($p = 0.71$). Then, we conducted a follow-up planned comparison between groups, combined across all three categories. Similar to previous studies, our VMF group (mean = 9.93%, sd = 6.65) made more cyclical choices than the HC group (mean = 5.71%, sd = 4.05; Wilcoxon ranked sums Z = 1.64, $p = 0.05$). The percentage of cyclical choices we observed were also roughly similar to those previously reported for these two groups[11,12].

However, we did not replicate that this increase in cyclical choices is selective to VMF damage in the frontal lobe. Unlike previous studies, our FC group (mean = 9.09%, sd = 3.74) also made more cyclical choices than the HC group (Z = 2.05, $p = 0.02$) and the difference between the VMF and FC groups was not significant (Z = 0.12, $p = 0.45$). This increase in cyclical choices in the FC group was unexpected, and we consider possible interpretations of this result in the discussion.

In addition to testing whether individuals with VMF damage exhibited more choice cycles as a group, we also examined the number of cyclical choices at the individual level. We did this to establish that our conclusions applied not only at the group level for individuals with VMF damage as whole, but also at the individual level for at least some specific individuals within that group (especially as the test of probabilistic transitivity is also performed at the level of individuals). To do this, we considered each individual with a VMF or other frontal lesion as a single case, and compared their percentage of choice cycles (across all three categories) against HCs. We made this comparison

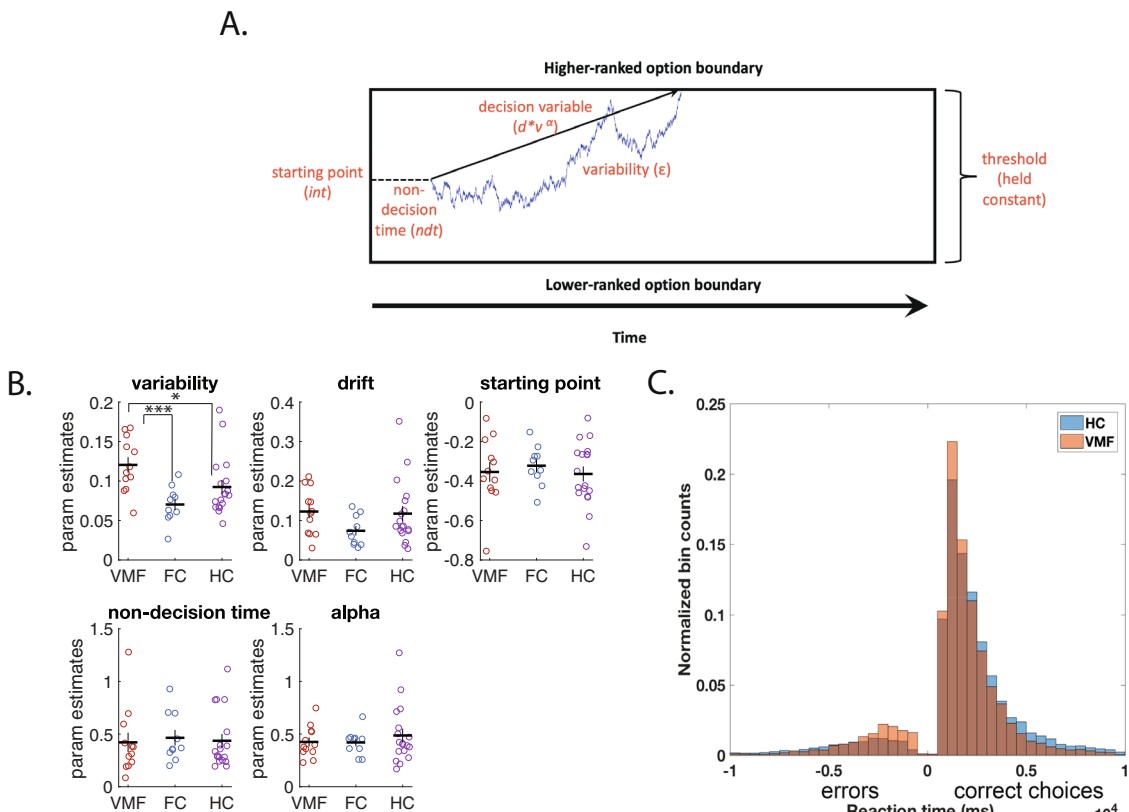

**Fig. 2 | Participants with VMF damage have higher decision variability.**
**A** illustration of the drift diffusion model. After the non-decision time (part of the reaction time not related to the decision process), the decision variable begins at a starting value and accumulates variable evidence at each time step towards one of the two choice options, until it reaches threshold for one of them. **B** DDM parameter fits: variability, drift rate, initial starting point, non-decision time, and alpha (exponent on rank distance). Error bars are standard errors of the mean. * denotes

$p = 0.047$, *** denotes $p < 0.001$, two-tailed $t$-tests. $N = 40$ individuals. VMF = Ventromedial Frontal group; FC = Frontal Controls; HC = Healthy Controls.
**C** Histogram of reaction times of all choices by the HC group (orange) and the VMF group (blue). RTs of "correct" choices, choosing the option with the higher average rank, are on the right, and RTs of "errors", choosing the option with the lower average rank, are mirrored on the left). Source data for **B** and **C** are provided as a Source Data file.

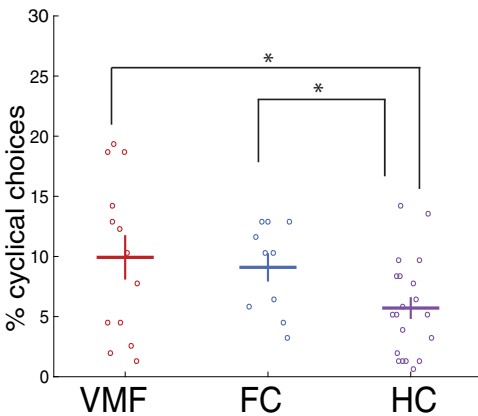

**Fig. 3 | Individuals with frontal damage exhibit more choice cycles.** Percentage of choice cycles across all domains. Error bars are standard errors of the mean. VMF Ventromedial Frontal group; FC Frontal Controls; HC Healthy Controls. * denotes $p = 0.05$ for VMF vs. HC, $p = 0.02$ for FC vs. HC, one-tailed Wilcoxon ranked sum tests. Source data are provided as a Source Data file.

using one-tailed case-control $t$-tests[22], which are modified to compare an individual against a normative group when the sample size is small. In the VMF group, four individuals made significantly more cyclical choices than HCs, before corrections for multiple comparisons (VMF2: t(19) = 2.04, $p = 0.03$; VMF8: t(19) = 3.28, $p = 0.003$, VMF10: t(19) = 3.13,

$p = 0.003$; VMF13: t(19) = 3.13, $p = 0.003$). These differences remained significant in the latter three individuals after correcting for multiple comparisons using FDR (corrected $p = 0.023$ for all three individuals). As described in the Supplemental Materials, we did not find that these three individuals differed from others in the VMF group in lesion location, lesion volume, or on any demographic variables. In contrast to the VMF group, none of the individuals in the FC group made significantly more cyclical choices than HCs (all $p >= 0.05$ before multiple comparison correction).

Finally, given that VMF subjects exhibited both increased choice cycles compared to HC subjects and a higher variability parameter in the DDM, we examined the correlation between these two measures. This correlation was significant in the VMF group (Spearman rho(10) = 0.67, $p = 0.02$), but was not significant across all subjects (rho(38) = 0.13, $p = 0.42$).

## Discussion
Past demonstrations that individuals with VMF damage more frequently make cyclical choices (i.e., selecting A over B, B over C, and C over A)[11–13]; these studies cannot distinguish between two possibilities, with drastically different implications for the function of the VMF. One possibility is that individuals with VMF damage have preferences that are fundamentally intransitive. A second possibility is that individuals with VMF damage have preferences that are more variable, yet still fundamentally transitive. Here we provide a clear test between these two possibilities by evaluating whether the choices of individuals with VMF damage satisfy a probabilistic model of transitivity. This model

assumes that choices are always generated according to a transitive preference ordering, but allows the specific ordering to vary from choice to choice. Choosers with variable preferences fit this model, but choosers with intransitive preferences do not. We find unambiguous evidence that individuals with VMF damage have fundamentally transitive preferences, even though their preferences are more variable, as all individuals with VMF damage make choices in all domains that are consistent with a probabilistic model of transitivity.

None of the individuals with VMF damage in our study violated the mixture model, even though their decisions exhibited more choice cycles than healthy controls. We interpret this pattern of results to mean that individuals with VMF damage have preferences that are more variable. That is, as suggested by[12], "values are unstable, fluctuating from trial to trial in those with VMF damage." We provide further support for this claim by fitting a DDM[20] to each individual's choices. In this model, the VMF group had significantly higher variability than healthy individuals or those with frontal damage outside the VMF. Importantly, the VMF group did not differ from others in any other parameter of the DDM, showing that preference variability was the only dimension affected by VMF damage.

These results, particularly that individuals with VMF damage have higher decision variability in the DDM, are broadly consistent with previous studies that have linked choice variability to variability in neural value signals in VMF[23–25] and that have shown that disruptions of VMF cause higher choice variability[26,27]. Many of these studies have modelled variability as arising from randomness in underlying values, such as in the DDM[23,28], and thus have assumed a probabilistic model of transitivity. However, such valuation models are not the only potential account of choice[15,29,30]. Our study takes a step beyond previous work by demonstrating that such valuation models are merited in our data, using a test of a particular probabilistic model of transitivity[31], the mixture model. The choices of all our subjects in the artworks category, and most of our subjects in the brands and gambles categories, were consistent with the mixture model. Combined with previous work testing this model[14], this suggests that human choices in a wide variety of domains are consistent with transitivity.

These results are difficult to reconcile with the view that the VMF is the only critical substrate for value-based choice[32]. This view would predict that individuals with VMF damage would only be able to choose in a non-value-based manner, for example, according to rules or heuristics. Rules and heuristics can approximate transitive preference orderings under some conditions[33], but are not generally guaranteed to do so[15]. In contrast, these results are easier to reconcile with a framework in which valuation and value-based choice are distributed processes, to which multiple regions of the brain contribute in some respect[34]. This framework would predict that other regions can compensate for damage to the VMF, so that VMF damage does not fundamentally abolish the transitivity of preferences. Because making transitive choices that maximize value is incredibly important to the survival of an organism, it would make sense that valuation is a highly conserved process that is robust to damage to one part of the cortex.

On this distributed view, there are several structures that may interact with the VMF to support value-based choice. Like the VMF, neural activity in the striatum reliably scales with subjective value across dozens of functional neuroimaging studies[2]. Whether striatal damage affects the variability of value-based decisions has not been studied to our knowledge, though striatal damage does impair value-based learning[35]. Similar to VMF damage, damage to the hippocampus also results in more variable or inconsistent decisions[36,37]. The hippocampus has been proposed to support deliberation about value, perhaps by retrieving evidence about value from memory[36,38]. The VMF is anatomically connected to, and functionally interacts with, both striatum and hippocampus, and so either or both networks may contribute to value-based choice[39,40]. Future studies could use functional

brain imaging to more directly test hypotheses about brain networks that may compensate in individuals with VMF damage.

Future studies could further investigate exactly how the VMF supports the stability and reduces the variability of preferences. One possibility is that VMF contributes to the computation of subjective value. If the subjective value is computed through the interaction of several brain regions, the loss of VMF may make this computation more variable and less reliable. An alternative possibility, though, is that the VMF contributes to the same preference ordering being repeated reliably, without contributing to valuation per se. For example, individuals might use episodic memories of their previous choices (e.g., "I remember choosing A over B before") to guide their decisions, or a representation of the context of the experiment may activate a specific set of preferences, as in a schematic network. Previous work has shown VMF involvement in both episodic memory processes[41] and schema formation[42,43].

Unexpectedly, we also found that the frontal control group made more cyclical choices than healthy controls. This finding is inconsistent with previous studies, where an increase in cyclical choices was specific to VMF damage in the frontal lobe[11,12]. One difference between our study and previous ones is that ours included the choice category of gambles. As the dorsomedial frontal, dorsolateral frontal, and insular cortices have been previously implicated in decisions about risk[44,45], the inclusion of choices between gambles could account for the discrepancy in results. Future studies should examine the generality and replicability of our findings regarding other frontal regions. We hesitate to draw strong conclusions about the role of these other frontal regions in preference variability based on our results alone. First, unlike individuals with VMF damage, the frontal controls did not differ from healthy controls in decision variability, or any other parameter, when their choices were fit to a DDM. Thus, the differences in the frontal control group were only observed in one of the two choice sets in our experiment (set B), whereas the differences in the VMF group were observed in both choice sets (sets A and B). Second, unlike individuals with VMF damage, the frontal controls only made significantly more cyclical choices at the group level; none of the individuals in the frontal control group made significantly more cyclical choices at the individual level when compared to healthy controls in a case-control test. Nonetheless, to the extent that damage to frontal regions outside the VMF also increases preference variability, this would further support the view that valuation is a distributed process in the brain.

In conclusion, we provided a clear-cut test of whether VMF is necessary for transitive preferences and rational choice and found that individuals with VMF damage have preferences that are more variable but, without exception, fundamentally transitive. This result clarifies how apparently erratic choices manifest after damage to the VMF[46,47] and potentially explains why studies using similar decision-making paradigms in individuals with VMF damage can yield different results[48]. Our findings further characterize the necessary role the VMF plays in value-based decision-making. Specifically, though each choice still reflects a subjective preference ordering after VMF damage, an intact VMF is necessary for these preference orderings to remain stable and reliable across time and contexts.

## Methods
### Experimental design

**Participants.** Fourteen individuals with focal damage to the frontal lobes were recruited from the Focal Lesion Database (FoLD) at the University of Pennsylvania, and ten were recruited from the Cognitive Neuroscience Research Registry at McGill University[49]. Individuals were eligible to participate if they had a lesion primarily affecting the frontal lobes. One individual was excluded due to incomplete data collection (the individual completed one session and was not able to be scheduled for the second). Fourteen females and 9 males were

included in the final sample. Participants were tested a minimum of 5 months after injury (median = 10.29 years, range: 5 months to 17.75 years).

Participants were divided into two groups a priori based on location of damage, assessed with MR or computed tomography images by a neurologist blind to task performance. The VMF group consisted of individuals who sustained damage to the VMF, which is defined as the medial wall below the genu of the corpus callosum, and the orbitofrontal cortices. The FC group consisted of individuals who sustained damage to the frontal lobe sparing the VMF, which includes damage to dorsolateral, dorsomedial, and insular cortices. Lesions were drawn on a common space [Montreal Neurological Institute (MNI) brain] by neurologists at the research sites blind to task performance. The overlap images for the groups are presented in Fig. 1. Damage in the VMF group was caused by aneurysm or subarachnoid hemorrhage in 5 cases, stroke in 2 cases, and tumor resection in 6 cases. Damage in the FC group was caused by hemorrhage, stroke, or infarct in 7 cases, and tumor resection in 3 cases.

Age and education matched HC individuals were recruited from the corresponding Normal Control Databases of the University of Pennsylvania ($N = 14$) and McGill University ($N = 6$), including 15 females and 5 males (Table 1). They were free of neurological and psychiatric disorders. All subjects provided informed consent and were compensated for their time. The study protocol was approved by the institutional review boards of both the University of Pennsylvania and McGill University.

**Apparatus.** All tasks were programmed using EPrime 2.0 (Psychology Software Tools). Participants were tested at the Hospital of the University of Pennsylvania, at the MNI, or at their own home in the greater Philadelphia or Montreal area. Participants saw stimuli on a laptop monitor and responded using the 1 and 0 keys of the keyboard.

**Items.** Choice items consisted of images of artwork, chocolate bars that differed in brand, and gambles presented as pie charts. There were two sets of items: 5 items from each of the categories (artworks, chocolate bar brands, gambles) that were used in repeated choices that allow a probabilistic test of transitivity (set A); and 10–11 items from each of the categories (10 for chocolate bar brands, 11 for art and gambles) used in non-repeated choices that allow testing for choice cycles (set B). Choices constructed using set A and set B stimuli were intermingled in each block. For both the artwork and chocolate bar brand categories, we designed item sets in which the items were normed to be close in preference. The gambles in the gamble category were all of equal expected value (Supplementary Materials).

**Procedure.** Participants completed a binary forced choice task. On each trial, participants first saw a central fixation point for 1 s, then a screen with two choice items (placed to the left and the right of the center). Participants indicated which item they preferred, by pressing buttons for left or right. Participants had as much time as they needed to make their selection. Following their selection, there was an intertrial interval of 1 s where a black screen was presented.

For set A stimuli, participants faced all possible pairings of 5 items, constituting 10 pairs, and each pair was repeated 15 times. The burden of prolonged testing, particularly on our older subjects, many of whom have brain injuries, limited us to using 15 repetitions per choice per person. This is smaller than the common rule of thumb of 20 for asymptotic tests of binomials. This smaller sample size may slightly bias those $p$-values not equal to 1 in the QTEST 2.1 results[50]. For set B stimuli, participants faced all possible pairings of either 10 (for brands) or 11 (for artworks and gambles) items, constituting 45 or 55 pairs in total, respectively, and each pair was presented once. Therefore, there were 195 (for brands) or 205 (for artworks and gambles) total choices in each category across the entire experiment.

Choice trials were presented in blocks, in which participants made choices between items within a single category (artworks, brands, gambles). There were five blocks of choices for each category, containing 39 (for brands) or 41 (for artworks and gambles) trials each. Each block contained 30 choices composed from set A and 9 or 11 choices composed from set B. Choices from set A and set B were intermingled with each other within a block, with the set B choices inserted into a block of A choices in positions randomly selected from a uniform distribution. We took several steps to reduce any potential memory effects for repeated set A choices. We designed the sequence of trials so that: (1) the same pairing was not repeated within a minimum of 3 trials; (2) the same item rarely appeared on immediately adjacent trials (no more than 9 times throughout the entire experiment); and (3) when the same pairing was repeated the choices immediately preceding and following that pairing differed from its previous occurrence (to minimize contextual memory). Furthermore, the side on which stimuli were presented was counterbalanced across repetitions. Finally, we divided the experiment into two sessions, held on separate days for every subject except two (due to scheduling constraints). The two sessions were held on average 8.09 (sd = 11.73) days apart (excepting the two who were tested on the same day, the sessions ranged from 1 day to 57 days apart). We did not observe a significant correlation between the total number of choice cycles and days between the two sessions (r(41) = 0.24, $p = 0.12$).

## Statistical analysis

**Tests of a probabilistic model of transitivity.** All data were analyzed with MATLAB (Mathworks). We used the set A choices to perform tests of a probabilistic model of transitivity. We first obtained the choice percentages (out of a possible total of 15 choices) for each of the 10 choice pairs (representing all possible pairings of the 5 items) in each category. We then tested the mixture model of preference described by Regenwetter and colleagues[14]. The mixture model states that a person's response comes from a probability distribution over all possible strict linear orderings of the items. Thus, preferences are transitive, but one's transitive state at any given time can vary. The probability of a person choosing one item (X) over another (Y) is the sum of all the preference states in which X is preferred to Y. In a two alternative forced choice task, this probability is constrained by the triangle inequalities. For every distinct X, Y, and Z in a choice set:

$$P_{xy} + P_{yz} - P_{xz} \leq 1 \qquad (1)$$

Where $P_{xy}$ denotes the probability of choosing X over Y, etc. For up to 5 options in a two alternative forced choice task, satisfying the triangle inequalities is necessary and sufficient for a set of choices to be consistent with the mixture model.

For choice percentages that did not satisfy the triangle inequalities, we used the QTEST 2.1[51,52] software to determine whether these violations were unlikely to be due to random sampling. QTEST 2.1 uses maximum likelihood estimation to find the goodness of fit of the data at each vertex in the linear ordering polytope defined by the triangle inequalities, using a chi-bar-square distribution with simulated weights[21,51]. A $p < 0.05$ was taken as evidence that a subject's choices in that category were inconsistent with the mixture model of preference.

**Drift diffusion modeling and analysis of reaction times.** We fit a DDM[20] to the choices and RTs from all set A choices for every subject and category in our experiment. We modeled the decision process as a decision variable (DV) that increased linearly with a slope $d*v^{\alpha}$, where $d$ was the drift rate, $v$ was the value difference between the items (expressed as the absolute rank difference between the two items for that individual), and $\alpha$ was an exponent accounting for potential nonlinearities in the effect of rank difference. We also assumed that at each time step there is Gaussian variability added to the DV, with a standard

deviation of $\varepsilon$. We assumed 10 ms time steps. We also assumed there is a non-decision time (ndt) before accumulation begins, and an initial value (int) of the DV that is constant across trials. Choices are made when the DV crosses a threshold.

Thus, there are five free parameters: $d$, $\alpha$, $\varepsilon$, int, and ndt. Note that the threshold was a fixed parameter across subjects, as one of the threshold, $d$, or $\varepsilon$ must be fixed for the other two parameters to be estimable. We chose to fix threshold after a model-comparison process showed that option to provide the best model fits. Threshold was held constant at (+/−) 0.15. Values for $d$ were sampled between 0 and 1, for $\varepsilon$ were sampled between 0 and 1, for $\alpha$ were sampled between 0 and 3, for int were sampled between the threshold bounds, and for ndt were sampled between 0 and the minimum RT minus 10 ms for that subject.

To fit these free parameters, we first calculated the cumulative probability that the DV crossed the threshold for the subject's choice ($T_{correct}$ or $T_{incorrect}$, where "correct" was defined as choosing the option of higher rank) across all time steps. For each trial, we then calculated the joint likelihood of the subject's choice at the time which they made that choice (their trial RT, minus ndt), by taking the derivative of this cumulative probability at the timestep of the subject's choice (every 10 ms to the maximum RT for the subject). The model was then fit using the MATLAB function *fmincon*, where the cost function was defined as the sum of the negative log likelihoods of the instantaneous probabilities of the subject's choices and RTs in all trials. The fitting procedure was repeated 10 times for each subject, with each iteration varying in randomly sampled starting values for the free parameters as specified above; the parameters with the lowest log likelihood out of the 10 was taken for that subject. The model was fit individually to each of the three categories (artworks, brands, gambles) for each subject.

We calculated the average ranks of the items according to the number of times each item was chosen, with the item that was chosen most often overall ranked first, the item chosen second-most ranked second, etc. We broke ties by looking at which item was more often chosen more than half of the time in every pair[12]. Three subjects still had tied ranks after this process, in one category each: two were HC subjects in the gambles domain, and the other was a VMF subject in the artworks domain. These subjects in these categories only were dropped from the DDM.

To look at differences in DDM parameters between groups across categories, we performed a mixed ANOVA on each of the free parameters, with group as the between-subject factor and item category as the within-subject factor.

**Choice cycles.** We used the set B choices to directly replicate previous studies that counted the number of choice cycles. We first determined the preference ordering within each category for each subject. The 10 or 11 options within each category were ranked according to how many times each was chosen by that subject. Then, for each trial, a choice was counted as cyclical if a lower-ranked item was chosen over a higher-ranked item. Following[12], ties were maintained in the rankings (i.e., more than one option could have the same rank) to provide a more conservative definition of cyclical choices. We used a 2-way mixed measures ANOVA of the number of choice cycles made by each participant, with group as a between-subjects factor and item category as a within-subject factor, followed by one-tailed Wilcoxon ranked sum post hoc pairwise tests when testing for effects between VMF and HC (as several previous studies have found increased cyclical choices after VMF damage, we had strong hypotheses about the direction of the results).

**Reporting summary**
Further information on research design is available in the Nature Research Reporting Summary linked to this article.

## Data availability
The dataset analyzed during the current study are available at the Center for Open Science repository at the following link: https://osf.io/cpwx2/. Source data are provided with this paper.

## Code availability
The code for the analyses central to the conclusions in this study are available at the Center for Open Science repository at the following link: https://osf.io/cpwx2/. The QTEST 2.1 package is available at the following link: http://regenwetterlab.org/qtest-2-1.

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

## Acknowledgements

We would like to thank Avinash Vaidya and Arthur Lee for constructive discussion on aspects of data analysis. We would also like to thank Lesley Fellows for facilitating access to participants in Montreal, Christine Déry and Eileen Cardillo for coordinating participants in Montreal and in Philadelphia, and all of the participants themselves, without whom this work would not be possible. This work was supported by National Institute on Drug Abuse (NIDA) R01-DA029149 to JWK and a Natural Sciences and Engineering Research Council (NSERC) postgraduate doctoral scholarship to LQY. QTEST 2.1 was developed with support by the National Science Foundation grants SES 10-62045 and SES 14-59699 (PI: M. Regenwetter) as well as by the Humboldt Foundation (Co-PIs: J. Stevens and M. Regenwetter).

## Author contributions

L.Q.Y, J.D., and J.W.K. conceptualized and designed the study. L.Q.Y. carried out the experimental procedures and performed the analyses with input from J.W.K. and J.D. All authors contributed to the writing of the manuscript.

## Competing interests

The authors declare no competing interests.
