## [Peer Review File · Nature Communications]

Individuals with ventromedial frontal damage display unstable but transitive preferences during decision makingREVIEWER COMMENTS

Reviewer #1 (Remarks to the Author):

Review of Nature Communications Manuscript titled
Individuals with ventromedial frontal damage have more unstable but still fundamentally transitive preferences.

This paper applies a modern and sophisticated approach to evaluating transitivity of preferences in patients with VMF damage, as well as in others. In so doing, it takes a very major step towards avoiding the many conceptual, mathematical, and statistical pitfalls in the old literature on transitivity. This provides an extremely valuable advance on a very difficult problem. The paper is very well motivated and fairly well crafted.

I strongly support publication, but with a major revision.

Main Comments:

On page 7, the paper claims that the “mixture model” was “developed by Regenwetter and colleagues (14).” This is wrong. They were the first to implement a correct statistical test on suitable data, but the model had been around since the 1960s at the very least. Maybe the authors should have another look at (14) for the history of the model.

Sample size: The study uses 15 observations, per choice pair, per person. A rule of thumb for safe reliance on asymptotic behavior (including in QTEST) is 20 observations per person, per choice pair. This is probably not a big deal, but the authors should acknowledge that their within-person sample size may slightly bias their p-values. (They could also consider adding Bayesian measures using the latest QTEST version. I am not insisting, just recommending.)

Data availability and result reproducibility: I am grateful that the paper made some of its data available in Table 2. In this day and age of open science, all the data should be made, both available, and accessible, easily (getting the data and understanding how they are coded should be painless). Data like these provide a great opportunity for others to contribute to the research endeavor.

I appreciate that the percentages in Table 2 let me attempt to reproduce some of the analysis results. I generated the results in the table below using QTEST 2.1. The main entries are frequentist p-values, the parenthetical values are approximate Bayes Factors (against an unconstrained model, where a value of, say >3 is good support in favor of the mixture model).

Art Brands Gambles

VMF2 .66 (11) 1 (13) .06 (6)

VMF8 1 (19) 1 (6) 1 (5)

VMF10 1 (2) 1 (1.4) 1 (5)

VMF12 1 (17) 1 (12) 1 (12)

My analysis of VMF2 does not align with the reported findings in the paper. This could have various causes: I may have reconstructed the data incorrectly from the percentages given in the paper. There could be typos in one of the tables in the paper. There might be other causes. I have appended my reconstructed data files and the order-constraints file for the mixture model, at the end of this report. The Bayesian analysis supports the paper’s findings (note that the “perfect” fit for VMF10 Brands only provides circumstantial evidence in Bayesian terms) for these data. In particular, combining the three stimulus sets, the group Bayes factors, which we obtain by taking the product of the three BFs for the three stimulus sets (for VMF2: > 800 ; for VMF8: 500; for VMF10: >10 ; for VMF12: > 2400), with the exception of VMF10 which is moderate, give extremely strong support for the mixture model jointly across stimulus sets.

Number of cyclical choices: The paper is extremely charitable with the nonsense statistic of the “number of cyclical choices.” It is well established that this is a completely meaningless statistic which measures nothing useful whatsoever. The main reason is that, in paradigms in which each choice option is paired against each other choice option, even flipping only one single pairwise choice of an individual can change the tally of “cyclical choices” by a small or a huge number. This means that “significantly more cyclical choices” (a term referred to, e.g., p.18) is a nonsense statement, because it is ill-defined what to call “significant” when one can make a statistic jump around hugely and arbitrarily with just a tiny tweak in the data (such as reversing one single data point). I could not really understand the section on “Choice Cycles,” it’s kind of cryptic. See also the next point.

The work on pages 23-24 is confusing in that it appears to suggest that the authors might have used the same data more than once for their analysis, thereby potentially undermining the validity of their statistical analysis. If one uses a set of data to create a hypothesis and then uses the same data to also test that hypothesis, then the statistical test is invalid because some unknown number of degrees of freedom have been used more than once.

Noise: The paper uses the term “noise” in a vague and rather misleading way. It would be very useful, in this context, to distinguish between overt (observable) choices and latent (unobservable) preferences. The paper does not look at noise in the sense of probabilistic response errors. I would therefore recommend dropping all reference to “noise.” The paper looks at variability/uncertainty of latent preferences that are all, and without exception, transitive. This is important, since the individuals with VMF damage are successfully modeled with probabilistic transitive preferences and no error in responses! It is much more powerful to conclude that VMF patients are more uncertain and more volatile in their preferences while being transitive, than to vaguely state that they are more “noisy.”

I would personally never submit a paper, especially to a prestigious journal, without careful and clean editing. The writing of this manuscript is rather sloppy, both in grammar and in style. Sloppy writing signals sloppy science. Had I been unaware of the mixture model’s great value for understanding transitivity, I might have recommended a rejection.

Minor comments:

Terminology: The mixture model is probabilistic, but not stochastic in the technical sense that the latter term is used in probability and statistics. This is because “stochastic” would require random variables indexed by time to model a process that unfolds in time. The drift diffusion model is stochastic, but the mixture model is not. Lay people often use the two terms as synonyms, but this is inaccurate. Certainly, the term “stochastic test of transitivity” used on page 5 is entirely nonsensical. The same applies to the term “axiomatic test” on page 16. That’s also nonsensical language.

Page 8: “only 5% of all possible choice percentages satisfy the triangle inequalities” is technically not wrong but it is misleading. It would be helpful to make clear that this statement is really about the data generating probabilities, and not percentages. The percentages should be subject to statistical inference, where we have to deal with “nonsignificant violations,” and everything quickly becomes very complicated and highly nuanced. It is also important to phrase things such that one does not give the false impression that the triangle inequalities might be sufficient in general. (Granted, for 5 choice options they do fully characterize the model.)

Page 8, “follow Tversky (15)’s lexicographic semiorder heuristic” Bold claim coming out of nowhere. How do you know this? Does that not also require fancy statistics?

QTEST should be written in all caps, with the first two letters larger in size. The rest of this document contains files I used for my QTEST analysis.

10 2 3 1

"VMF2 Art"

14 1

15 0

15 0

14 1

10 5

12 3

10 5

5 10

12 3

7 8

"VMF2 Brands"

10 5

9 6

11 4

12 3

6 9

13 2

13 2

9 6

11 4

11 4

"VMF2 Gambles"

15 0

13 2

15 0

14 1

14 1

15 0

15 0

11 4

15 0

13 2

10 2 3 1

"VMF8 Art"

10 5

12 3

15 0

15 0

6 9

15 0

14 1

12 3

15 0

10 5

"VMF8 Brands"

9 6

14 1

11 4
12 3
12 3
10 5
14 1
12 3
12 3
6 9

"VMF8 Gambles"

14 1
15 0
14 1
15 0
14 1
13 2
15 0
14 1
14 1
14 1

10 2 3 1

"VMF10 Art"

15 0
15 0
15 0
15 0
15 0
15 0
15 0
14 1
15 0
15 0

"VMF10 Brands"

15 0
15 0
15 0
15 0
15 0
15 0
15 0
15 0
15 0
13 2

"VMF10 Gambles"

14 1
15 0
14 1
15 0
14 1
13 2
15 0
14 1

14 1
14 1

10 2 3 1

"VMF12 Art"

9 6
8 7
10 5
9 6
8 7
11 4
10 5
10 5
11 4
8 7

"VMF12 Brands"

9 6
9 6
9 6
10 5
7 8
12 3
8 7
7 8
10 5
9 6

"VMF12 Gambles"

6 9
13 2
11 4
14 1
11 4
10 5
10 5
11 4
11 4
9 6

Order-constraints file for the mixture model:

40 10
-1 0 0 0 0 0 0 0 0 0
0 -1 0 0 0 0 0 0 0 0
0 0 -1 0 0 0 0 0 0 0
0 0 0 -1 0 0 0 0 0 0
0 0 0 0 -1 0 0 0 0 0
0 0 0 0 0 -1 0 0 0 0
0 0 0 0 0 0 -1 0 0 0
0 0 0 0 0 0 0 -1 0 0
0 0 0 0 0 0 0 0 -1 0
0 0 0 0 0 0 0 0 0 -1 0

1
1
1
1
1
1
1
1
1
1
1
1
1
1
1
1
1
1
1
1

Reviewer #2 (Remarks to the Author):

Yu and colleagues examined if people with VMF lesion, compared to people with lesions in other frontal brain regions, and healthy controls, display consistent choices. They conducted a decision-making task across three different categories – artwork, chocolate bars, and monetary gambles. Using several analysis approaches, they managed to disentangle two possible sources for subjects' seemingly inconsistent choices - actual intransitive preferences (inconsistent choices) and variability in their choices. The authors show that people with lesioned VMF display rational decisions consistent with transitive preferences, but they exhibit greater variability in their preferences.

I think that the study is very interesting and important. It is the first time that the seemingly inconsistent choices that were previously observed in people with damage to the VMF is attributed to variability in choices and not actual transitive preferences. This finding helps to pinpoint the exact function of the VMF in value-based decision making and also demonstrates, using a previously established method, how to disentangle between transitive preferences and noisiness in the decision process.

I do have several comments that need to be addressed, but I think that if the authors will adequately address my comments, this manuscript is suitable for publication in Nature Communications.

1) Usually, in a well powered model of choice consistency (see Bronars (1987) for a discussion of it), like in a GARP test using sufficient number of budget sets, (see Choi et al., (2007)) and when using Afriat index or Varian index, there is a relatively large variance across subjects in their consistency scores, and rarely one will find perfect subjects (unless they always choose in the corners). However, because in the current study most subjects are perfect, I wonder how much power there is in the design or in the analysis method. If almost all subjects demonstrate perfect consistency, then this raises the possibility that the study design was under powered. Therefore, the conclusion might be due to lack of power and not because lesioned subjects truly have transitive preferences.

2) I think that it will be interesting to examine if there is a correlation between the number of choice cycles each subject had and any of the DDM parameters (especially the noise parameter)? It might be the case that a significant correlation will hint for some relationship between the two features. A lack of correlation will hint that these processes are not related and do not influence each other.

3) The authors demonstrated that at the group level subjects with VMF lesion have more choice cycles than healthy controls but that FC subjects also have more choice cycles. This raises an important point. As the authors demonstrate, when they examine the data on a subject basis, they show that only 3 subjects from the VMF group have more choice cycles and they are not different from the FC group. Hence, the conclusion about the VMF group is based only on 3 subjects. So, I think that the authors need to tone down their general claim about the unique role of the VMF compared to other frontal regions in consistent choices. Importantly, this is not the case in the noise parameter of the DDM, where there is a much stronger specific effect for the VMF group (and at the individual level).

4) The point I raised in 3, raises a related concern. How can we interpret the lack of difference in the number of choice cycles between the FC and VMF groups? This is a crucial point, because if there is a connection between structure and function, between lesion and behavior, then the group conclusion is problematic as a whole. The differences (or lack of them) either do not arise from the lesion location per se or they do, but then it originates from some other attribute of the lesion that the authors can't identify. I agree that this might be because a lack of power (not enough subjects, or again the task is underpowered to pick up small differences in behavior) but still this is a key issue that needs to be addressed. I know this is probably very hard to do, but finding any evidence that relates structure and function will strengthen the authors claim.

5) Because of these issues, I think that the authors also need to restate some of their claims in the discussion. For example, as they conclude their main claim of the paper, (lines 300-302), I think they should address the possibility that they did not find subjects with intransitive preferences because of low power in the design. Furthermore, in line 305, they claim that at the group level VMF subjects show more choice cycles. However, because this is driven by only 3 subjects, I do not think that they have a strong claim at the group level for choice cycles resulting by lesions specific to the VMF.

Reviewer #3 (Remarks to the Author):

This is an excellent paper that describes a well-designed study with results that add an important contribution to the literature. I have very few comments.

Summary.

There is an extensive literature describing the role of the vmPFC in value-based decisions. In recent years, the bulk of the studies have used single cell recordings or functional MRI, measuring how changes to neural or BOLD activity correspond to subjective value and value-based choices. However, these sorts of results have not been able to provide a clear cut view of the role of the vmPFC.

Here, the authors add a critical test of the role of the vmPFC in value-based decisions by observing choices of patients with vmPFC. This work adds to a much smaller but highly important set of studies which have examined patterns of choices in patients with brain lesions. Critically, the current study extends previous work by addressing a pointed question about transitivity in patients with vmPFC with a beautifully designed study and a new approach to analyzing the data. The results provide clear-cut results showing that patients with vmPFC damage are noisier in their choices, but are not intransitive.

Comments.

These results have important implications for theories of vmPFC function and for neural substrates of value-based decisions more generally. I was particularly intrigued with the speculation that the results may support a more distributed view of value-based decisions and involving multiple neural structures and their interaction. However, it seemed to me that this point could be expanded to (a) discuss which other structures may be involved and (b) what might be the specific role of vmPFC vs. other structures. In particular, the authors suggest that individual may use episodic memory of previous choices; is the suggestion that there may be involvement of interaction between the vmPFC and the hippocampus, in this regard? This idea would seem to fit well with recent experimental findings and theories (E.g. a recent study that examined value-based choices and DDM fits in patients with damage to the hippocampus; Bakkour, Palombo, et al., 2019. The hippocampus supports deliberation during value-based decisions. *elife.*, 8.)

Response letter

We thank the reviewers for the helpful and thoughtful comments. We have responded to their concerns below (our responses are in italic, with amended text of the manuscript highlighted yellow).

Reviewer One:

This paper applies a modern and sophisticated approach to evaluating transitivity of preferences in patients with VMF damage, as well as in others. In so doing, it takes a very major step towards avoiding the many conceptual, mathematical, and statistical pitfalls in the old literature on transitivity. This provides an extremely valuable advance on a very difficult problem. The paper is very well motivated and fairly well crafted.

I strongly support publication, but with a major revision.

Main Comments:

On page 7, the paper claims that the “mixture model” was “developed by Regenwetter and colleagues (14).” This is wrong. They were the first to implement a correct statistical test on suitable- data, but the model had been around since the 1960s at the very least. Maybe the authors should have another look at (14) for the history of the model.

Thank you for pointing this out. We have changed our language around this point and included references to previous work on the mixture model.

Text from page 7, line 159-162:

The mixture model (14, 18, 19) assumes that every choice is made according to a preference ordering, but that the preference ordering governing a specific choice is drawn randomly from a mixture of all possible preference orderings.

Sample size: The study uses 15 observations, per choice pair, per person. A rule of thumb for safe reliance on asymptotic behavior (including in QTEST) is 20 observations per person, per choice pair. This is probably not a big deal, but the authors should acknowledge that their within-person sample size may slightly bias their p-values. (They could also consider adding Bayesian measures using the latest QTEST version. I am not insisting, just recommending.)

Unfortunately, the testing burden on this population, given that they are all older and many have brain injuries, limited us to 15 repetitions per choice per person instead of 20. We have added the following statement in the Methods section acknowledging this limitation.

Text from pages 21, line 433-435:

The burden of prolonged testing, particularly on our older subjects, many of whom have brain injuries, limited us to using 15 repetitions per choice per person. This is smaller than the common rule of thumb of 20 for asymptotic tests of binomials. This smaller sample size may slightly bias those p-values not equal to 1 in the QTEST results.

As the reviewer suggested, we also computed Bayes factors using QTEST 2.1 and the procedure described in the associated paper (Zwilling et al., 2019, JMP). We agree with Reviewer 1 that the Bayesian analysis strongly supports our findings. However, we had some trouble getting some BFs to converge within a reasonable computational time. Gibbs sample numbers in the millions still did not lead to convergence for these subjects. The convergence difficulty was not related to lesion group status (in fact, the subjects for whom we had trouble reaching convergence were in the healthy control group), and appeared to have also been a factor for some subjects in Zwilling et al., 2019. Given this difficulty, and also considering the interpretive difficulties for a general audience (Reviewer 1 notes that there are perfect fits which provide only circumstantial evidence according the BF), we opted to discuss only the frequentist p-values.

Data availability and result reproducibility: I am grateful that the paper made some of its data available in Table 2. In this day and age of open science, all the data should be made, both available, and accessible, easily (getting the data and understanding how they are coded should be painless). Data like these provide a great opportunity for others to contribute to the research endeavor.

We agree that data accessibility is important. To this end, we have made the entire choice dataset and analysis scripts available at the following link:
<https://osf.io/cpwx2/>

I appreciate that the percentages in Table 2 let me attempt to reproduce some of the analysis results. I generated the results in the table below using QTEST 2.1. The main entries are frequentist p-values, the parenthetical values are approximate Bayes Factors (against an unconstrained model, where a value of, say >3 is good support in favor of the mixture model).

Art Brands Gambles
VMF2 .66 (11) 1 (13) .06 (6)
VMF8 1 (19) 1 (6) 1 (5)
VMF10 1 (2) 1 (1.4) 1 (5)
VMF12 1 (17) 1 (12) 1 (12)

My analysis of VMF2 does not align with the reported findings in the paper. This could have various causes: I may have reconstructed the data incorrectly from the percentages given in the paper. There could be typos in one of the tables in the paper.

There might be other causes. I have appended my reconstructed data files and the order-constraints file for the mixture model, at the end of this report. The Bayesian analysis supports the paper's findings (note that the "perfect" fit for VMF10 Brands only provides circumstantial evidence in Bayesian terms) for these data. In particular, combining the three stimulus sets, the group Bayes factors, which we obtain by taking the product of the three BFs for the three stimulus sets (for VMF2: > 800; for VMF8: 500; for VMF10: >10; for VMF12: > 2400), with the exception of VMF10 which is moderate, give extremely strong support for the mixture model jointly across stimulus sets.

We are grateful to the reviewer for the careful re-analysis of our data and for including the QTEST files. We took the possible discrepancy the reviewer found very seriously and investigated the matter thoroughly. The reconstructed files were correct. Our order-constraints file was different, but should be equivalent to the reviewer's, though we now use the order-constraints file provided by the reviewer. The discrepancy arose from differing versions of QTEST used. When our analysis was originally performed, we used the original version of QTEST. The reviewer used QTEST 2.1. Using the reviewer's order-constraints file in both versions, we were able to closely match the reviewer's p-values with QTEST 2.1, but obtained close to our original p-values using the original version of QTEST. We have now redone every analysis using QTEST 2.1 and report the results in Table 3. While some of the p-values have changed from the numbers produced by the original QTEST in the previous version of our paper, none of the outcomes were altered in the VMF group. A few outcomes were altered in the control groups. In the frontal control group, one person violated the mixture model in the gambles category with QTEST 2.1 who did not do so in the previous analysis, and so did 2 additional people in the healthy control group (making it 4 HCs who violated the model in total). None of these results changes our conclusion that VMF damage does not cause violations of transitivity according to the mixture model.

We have updated Table 3 and the associated paragraph describing the results (pg 8, lines 167-174):

Across the individuals tested, most choices – including all of those from individuals with VMF damage – were consistent with this probabilistic model of transitivity (124 of 129 total tests across all individuals and domains, **Table 3**). None of the individuals with VMF damage significantly violated the mixture model in any of the three domains (out of a total of 39 tests, **Table 3**). Only one of the individuals in the FC group significantly violated the mixture model in the gamble domain (out of a total of 30 tests, **Table 3**). Four individuals in the HC group significantly violated the mixture model in the gambles domain, and one in the brands domain (out of a total of 60 tests, **Table 3**).

In investigating this issue, we re-checked Tables 2 and 3 and did discover two typos in the original Table 2. VMF13 was listed as VMF12 and VMF11 was listed as VMF10. These errors have now been fixed in the new Table 2. They did not cause any apparent discrepancies in the reviewer's analyses because all four subjects (VMF10-

VMF13) satisfied the triangle inequalities in all three domains, and therefore yielded p-values of 1 in QTEST. We thank the reviewer for prompting us to discover and correct these.

Number of cyclical choices: The paper is extremely charitable with the nonsense statistic of the “number of cyclical choices.” It is well established that this is a completely meaningless statistic which measures nothing useful whatsoever. The main reason is that, in paradigms in which each choice option is paired against each other choice option, even flipping only one single pairwise choice of an individual can change the tally of “cyclical choices” by a small or a huge number. This means that “significantly more cyclical choices” (a term referred to, e.g., p.18) is a nonsense statement, because it is ill-defined what to call “significant” when one can make a statistic jump around hugely and arbitrarily with just a tiny tweak in the data (such as reversing one single data point). I could not really understand the section on “Choice Cycles,” it’s kind of cryptic. See also the next point.

We have included choice cycles because they were used as measures of transitivity in these past papers. We do not agree with their conclusions or their approach to testing transitivity, but we did want to show that we get the same results (for the VMF group against HC) using the same analysis as in previous studies. The reason we do so is to allay the concern that our individuals with VMF damage are somehow very different than those in previous studies.

We introduce the mixture model as a better way of answering the question of whether such individuals are truly intransitive, suggesting that previous findings of regarding choice cycles were actually attributable to higher variability. We include both measures in this paper for completeness and for relatability to previous findings. We have tried to be more careful not to convey that we agree with the choice cycles approach.

We have amended the text on page 12 (line 238-241) to the following in order to make it more clear that choice cycles are not sufficient to assess variability in choice:

Although we do not endorse counting cycles as a measure of transitive choice, given the problems identified with this measure (14, 21), such a replication would provide evidence that our individuals with VMF damage are not behaving differently than individuals with VMF damage in prior studies.

The work on pages 23-24 is confusing in that it appears to suggest that the authors might have used the same data more than once for their analysis, thereby potentially undermining the validity of their statistical analysis. If one uses a set of data to create a hypothesis and then uses the same data to also test that hypothesis, then the statistical test is invalid because some unknown number of degrees of freedom have been used more than once.

Each participant’s data can be separated into two distinct sets of choices. One of these (set A) consists of repeated choices, employed both for the mixture model and for

the DDM modelling. The mixture model and the DDM analyses address different questions with respect to the data, so there is no “double-dipping”. The mixture model establishes that individuals with VMF damage adhere to probabilistic transitivity. The DDM modeling tests the hypothesis that these individuals’ choices are more variable compared to the other groups. The other set (set B) of choices are used to calculate choice cycles, as a replication of previous papers.

Noise: The paper uses the term “noise” in a vague and rather misleading way. It would be very useful, in this context, to distinguish between overt (observable) choices and latent (unobservable) preferences. The paper does not look at noise in the sense of probabilistic response errors. I would therefore recommend dropping all reference to “noise.” The paper looks at variability/uncertainty of latent preferences that are all, and without exception, transitive. This is important, since the individuals with VMF damage are successfully modeled with probabilistic transitive preferences and no error in responses! It is much more powerful to conclude that VMF patients are more uncertain and more volatile in their preferences while being transitive, than to vaguely state that they are more “noisy.”

We agree with the reviewer that variability is a more precise descriptor than “noise” and have revised the manuscript accordingly.

e.g., changed text from page 16, lines 311-314:

These results, particularly that individuals with VMF damage have higher decision variability in the DDM, are broadly consistent with previous studies that have linked choice variability to variability in neural value signals in VMF (22-24) and that have shown that disruptions of VMF cause higher choice variability (25, 26).

I would personally never submit a paper, especially to a prestigious journal, without careful and clean editing. The writing of this manuscript is rather sloppy, both in grammar and in style. Sloppy writing signals sloppy science. Had I been unaware of the mixture model’s great value for understanding transitivity, I might have recommended a rejection.

We have given the manuscript a thorough readthrough for style and grammar. Thank you for your suggestions.

Minor comments:

Terminology: The mixture model is probabilistic, but not stochastic in the technical sense that the latter term is used in probability and statistics. This is because “stochastic” would require random variables indexed by time to model a process that unfolds in time. The drift diffusion model is stochastic, but the mixture model is not. Lay people often use the two terms as synonyms, but this is inaccurate. Certainly, the term

“stochastic test of transitivity” used on page 5 is entirely nonsensical. The same applies to the term “axiomatic test” on page 16. That’s also nonsensical language.

Thanks for the helpful clarification. We have replaced the term “stochastic” with “probabilistic” as it relates to the mixture model throughout the paper, and removed “axiomatic test” on page 16.

Page 8: “only 5% of all possible choice percentages satisfy the triangle inequalities” is technically not wrong but it is misleading. It would be helpful to make clear that this statement is really about the data generating probabilities, and not percentages. The percentages should be subject to statistical inference, where we have to deal with “nonsignificant violations,” and everything quickly becomes very complicated and highly nuanced. It is also important to phrase things such that one does not give the false impression that the triangle inequalities might be sufficient in general. (Granted, for 5 choice options they do fully characterize the model.)

Thanks for this point. We have changed the sentence on page 8 (line 164-166) so that it is hopefully less misleading on both issues:

For five stimuli and thus ten pairwise choice percentages, the triangle inequalities fully characterize the mixture model and impose rather restrictive constraints; only 5% of the sample space satisfies the triangle inequalities.

Page 8, “follow Tversky (15)’s lexicographic semiorder heuristic” Bold claim coming out of nowhere. How do you know this? Does that not also require fancy statistics?

Since in the new QTEST 2.1 analyses more individuals in the control groups (including one subject in the brands domain) significantly violate the mixture model, we have removed this statement.

QTEST should be written in all caps, with the first two letters larger in size. The rest of this document contains files I used for my QTEST analysis.

We have changed all mentions to QTEST to the format you suggest.

10 2 3 1

"VMF2 Art"

14 1

15 0

15 0

14 1

10 5

12 3
10 5
5 10
12 3
7 8

"VMF2 Brands"

10 5
9 6
11 4
12 3
6 9
13 2
13 2
9 6
11 4
11 4

"VMF2 Gambles"

15 0
13 2
15 0
14 1
14 1
15 0
15 0
11 4
15 0
13 2

10 2 3 1

"VMF8 Art"

10 5
12 3
15 0
15 0
6 9
15 0
14 1
12 3
15 0
10 5

"VMF8 Brands"

9 6

14 1
11 4
12 3
12 3
10 5
14 1
12 3
12 3
6 9

"VMF8 Gambles"

14 1
15 0
14 1
15 0
14 1
13 2
15 0
14 1
14 1
14 1

10 2 3 1

"VMF10 Art"

15 0
15 0
15 0
15 0
15 0
15 0
15 0
14 1
15 0
15 0

"VMF10 Brands"

15 0
15 0
15 0
15 0
15 0
15 0
15 0
15 0

13 2

"VMF10 Gambles"

14 1

15 0

14 1

15 0

14 1

13 2

15 0

14 1

14 1

14 1

10 2 3 1

"VMF12 Art"

9 6

8 7

10 5

9 6

8 7

11 4

10 5

10 5

11 4

8 7

"VMF12 Brands"

9 6

9 6

9 6

10 5

7 8

12 3

8 7

7 8

10 5

9 6

"VMF12 Gambles"

6 9

13 2

11 4

14 1
11 4
10 5
10 5
11 4
11 4
9 6

Order-constraints file for the mixture model:

40 10
-1 0 0 0 0 0 0 0 0 0
0 -1 0 0 0 0 0 0 0 0
0 0 -1 0 0 0 0 0 0 0
0 0 0 -1 0 0 0 0 0 0
0 0 0 0 -1 0 0 0 0 0
0 0 0 0 0 -1 0 0 0 0
0 0 0 0 0 0 -1 0 0 0
0 0 0 0 0 0 0 -1 0 0
0 0 0 0 0 0 0 0 -1 0
0 0 0 0 0 0 0 0 0 -1
-1 0 0 1 0 0 -1 0 0 0
-1 0 1 0 0 -1 0 0 0 0
-1 1 0 0 -1 0 0 0 0 0
0 -1 0 1 0 0 0 0 -1 0
0 -1 1 0 0 0 0 -1 0 0
0 0 -1 1 0 0 0 0 0 -1
0 0 0 0 -1 0 1 0 -1 0
0 0 0 0 -1 1 0 -1 0 0
0 0 0 0 0 -1 1 0 0 -1
0 0 0 0 0 0 0 -1 1 -1
0 0 0 0 0 0 0 0 0 1
0 0 0 0 0 0 0 0 0 1
0 0 0 0 0 0 0 0 1 0
0 0 0 0 0 0 1 0 0 0
0 0 0 0 0 1 0 0 0 0
0 0 0 0 1 0 0 0 0 0
0 0 0 1 0 0 0 0 0 0
0 0 1 0 0 0 0 0 0 0
0 1 0 0 0 0 0 0 0 0
1 0 0 0 0 0 0 0 0 0
0 0 0 0 0 0 0 1 -1 1
0 0 0 0 0 1 -1 0 0 1
0 0 0 0 1 -1 0 1 0 0
0 0 0 0 1 0 -1 0 1 0

Reviewer two:

Yu and colleagues examined if people with VMF lesion, compared to people with lesions in other frontal brain regions, and healthy controls, display consistent choices. They conducted a decision-making task across three different categories – artwork, chocolate bars, and monetary gambles. Using several analysis approaches, they managed to disentangle two possible sources for subjects' seemingly inconsistent choices - actual intransitive preferences (inconsistent choices) and variability in their choices. The authors show that people with lesioned VMF display rational decisions consistent with transitive preferences, but they exhibit greater variability in their preferences.

I think that the study is very interesting and important. It is the first time that the seemingly inconsistent choices that were previously observed in people with damage to the VMF is attributed to variability in choices and not actual transitive preferences. This finding helps to pinpoint the exact function of the VMF in value-based decision making and also demonstrates, using a previously established method, how to disentangle between transitive preferences and noisiness in the decision process.

I do have several comments that need to be addressed, but I think that if the authors will adequately address my comments, this manuscript is suitable for publication in Nature Communications.

1) Usually, in a well powered model of choice consistency (see Bronars (1987) for a discussion of it), like in a GARP test using sufficient number of budget sets, (see Choi et al., (2007)) and when using Afriat index or Varian index, there is a relatively large variance across subjects in their consistency scores, and rarely one will find perfect subjects (unless they always choose in the corners). However, because in the current study most subjects are perfect, I wonder how much power there is in the design or in the analysis method. If almost all subjects demonstrate perfect consistency, then this raises the possibility that the study design was under powered. Therefore, the conclusion might be due to lack of power and not because lesioned subjects truly have transitive preferences.

There is a critical difference between the tests of choice consistency the reviewer mentions and our approach. Tests of GARP, or Afriat's or Varian's index, assume there is no variability in the underlying preferences. If there is any variability, this would register as a violation or a less than perfect measure of consistency. In contrast, the mixture model that we evaluate allows for variability in the underlying preferences. That is, unlike those other approaches, the mixture model is a probabilistic test of transitivity. This is almost certainly the critical difference. If you consider the analysis of choice cycles in Figure 3, whether subjects exhibit any choice cycles at all would be a test of consistency that assumes no variability in the underlying preferences, and the vast majority of subjects do have a non-zero number of cyclical choices.

Nonetheless, power is a reasonable concern. We believe we have enough power as we are close to the recommended number of choices (15 repeated choices instead of the recommended 20, a reduction implemented to make testing less onerous on our older subjects with brain injuries). From the reanalysis with an updated version of QTEST, 5 subjects (1 in FC, 4 in HC group) out of 43 total significantly violated the mixture model in at least one category. These results are in line with what can be expected from (healthy) human subject data in Regenwetter et al. 2011 (where 1/16 subjects committed significant violations) – note that our subjects are significantly older than the undergraduate sample they tested and we now use an updated version of QTEST. These comparisons suggest that our test was capable of detecting violations, but that individuals with VMF damage did not violate the mixture model.

We have added a caveat about power in our methods section – reproduced below.

Text from pages 21, line 433-435:

The burden of prolonged testing, particularly on our older subjects, many of whom have brain injuries, limited us to using 15 repetitions per choice per person. This is smaller than the common rule of thumb of 20 for asymptotic tests of binomials. This smaller sample size may slightly bias those p-values not equal to 1 in the QTEST results.

2) I think that it will be interesting to examine if there is a correlation between the number of choice cycles each subject had and any of the DDM parameters (especially the noise parameter)? It might be the case that a significant correlation will hint for some relationship between the two features. A lack of correlation will hint that these processes are not related and do not influence each other.

Thank you for this suggestion. Keeping in mind the weaknesses of counting choice cycles as an approach (see Reviewer 1's comments), we examined the correlations suggested by the reviewer. In the VMF group, the variability parameter in the DDM (formerly "noise") is correlated with the total number of choice cycles ($\rho = 0.67$, see blue line in figure below), hinting at a relationship between these two features, though this relationship is not significant across all participants ($\rho = 0.13$, see black line in figure below). We have added the following two sentences in the results section to describe this relationship:

Page 14, line 279-282:

Finally, given the finding that VMF subjects exhibited both increased choice cycles compared to HC subjects and a higher variability parameter in the DDM, we examined the correlation between these two measures. This correlation was significant in the VMF group (Spearman $\rho(10) = 0.67$, $p = 0.02$), but was not significant across all subjects ($\rho(38) = 0.13$, $p = 0.42$).

3) The authors demonstrated that at the group level subjects with VMF lesion have more choice cycles than healthy controls but that FC subjects also have more choice cycles. This raises an important point. As the authors demonstrate, when they examine the data on a subject basis, they show that only 3 subjects from the VMF group have more choice cycles and they are not different from the FC group. Hence, the conclusion about the VMF group is based only on 3 subjects. So, I think that the authors need to tone down their general claim about the unique role of the VMF compared to other frontal regions in consistent choices. Importantly, this is not the case in the noise parameter of the DDM, where there is a much stronger specific effect for the VMF group (and at the individual level).

4) The point I raised in 3, raises a related concern. How can we interpret the lack of difference in the number of choice cycles between the FC and VMF groups? This is a crucial point, because if there is a connection between structure and function, between lesion and behavior, then the group conclusion is problematic as a whole. The differences (or lack of them) either do not arise from the lesion location per se or they do, but then it originates from some other attribute of the lesion that the authors can't identify. I agree that this might be because a lack of power (not enough subjects, or again the task is underpowered to pick up small differences in behavior) but still this is a key issue that needs to be addressed. I know this is probably very hard to do, but finding any evidence that relates structure and function will strengthen the authors claim.

5) Because of these issues, I think that the authors also need to restate some of their claims in the discussion. For example, as they conclude their main claim of the paper,

(lines 300-302), I think they should address the possibility that they did not find subjects with intransitive preferences because of low power in the design. Furthermore, in line 305, they claim that at the group level VMF subjects show more choice cycles. However, because this is driven by only 3 subjects, I do not think that they have a strong claim at the group level for choice cycles resulting by lesions specific to the VMF.

As the above three points are interrelated, we will address them together in this response.

First, we note that all of these concerns regard the analysis of choice cycles. As Reviewer 1 stresses, this is an imperfect measure, and our main conclusions do not hinge on it. The tests of the mixture model establish that individuals with VMF damage have transitive preferences. The DDM results establish that individuals with VMF damage exhibit higher variability in their preferences, compared to both the frontal control and healthy control groups. These two results support our main conclusion that individuals with VMF damage have more variable, yet still transitive, preferences.

The choice cycle analysis serves primarily to establish that the individuals we tested with VMF damage perform similarly on the same measure reported in previous studies of consistency. We agree completely that the increase in choice cycles we observe is not specific to VMF damage, as we also observed increased choice cycles after FC damage. We have amended the sentence the reviewer referred to on line 305 (now line 302) to make it clear that the VMF group exhibits more choice cycles than the healthy controls only.

We disagree, though, with the reviewer's claim that "the conclusion about the VMF group is based only on 3 subjects" or that "this [effect] is driven by only 3 subjects." The case-control t-test is not a test of whether an individual is an outlier within the VMF group. Rather, this is a test of whether we can conclude that that individual differs from healthy controls. Concluding that three subjects differ at the individual level from healthy controls does not somehow invalidate the test demonstrating a difference between the VMF subjects as a group and the healthy controls as a group. Consider, for example, two normal distributions representing two groups (test and control). As we increase the mean of test distribution, relative to the control distribution, we would observe that a larger percentage of the test distribution was significantly different from the control distribution according to the case-control test. But we would not want to take this as evidence against a difference in means between the two distributions; in the fact, this difference in means is also increasing. In our case, we observe both that the mean of the VMF group is larger than the healthy controls and that the mean for three individuals within this group is larger.

As stated above, we believe that while power is slightly lower in our design (and have added a caveat to this effect in revised manuscript), we still have performed a sensitive test of transitivity (demonstrated, for example, by the violations observed in the control groups). Power in terms of subjects is a frequent concern in lesion studies, due to the difficulties recruiting these rare subjects. Our group sizes are comparable to other similar group-based lesion studies (eg. Vaidya & Fellows, 2015, Nature Communications), including the papers that found significantly elevated choice cycles in VMF-lesioned subjects (Henri-Bhargava et al., 2012; Fellows and Farah, 2007).

Sentence from page 15, line 302:

None of the individuals with VMF damage in our study violated the mixture model, even though their decisions exhibited more choice cycles than healthy controls.

Reviewer #3 (Remarks to the Author):

This is an excellent paper that describes a well-designed study with results that add an important contribution to the literature. I have very few comments.

Summary.

There is an extensive literature describing the role of the vmPFC in value-based decisions. In recent years, the bulk of the studies have used single cell recordings or functional MRI, measuring how changes to neural or BOLD activity correspond to subjective value and value-based choices. However, these sorts of results have not been able to provide a clear cut view of the role of the vmPFC.

Here, the authors add a critical test of the role of the vmPFC in value-based decisions by observing choices of patients with vmPFC. This work adds to a much smaller but highly important set of studies which have examined patterns of choices in patients with brain lesions. Critically, the current study extends previous work by addressing a pointed question about transitivity in patients with vmPFC with a beautifully designed study and a new approach to analyzing the data. The results provide clear-cut results showing that patients with vmPFC damage are noisier in their choices, but are not intransitive.

Comments.

These results have important implications for theories of vmPFC function and for neural substrates of value-based decisions more generally. I was particularly intrigued with the speculation that the results may support a more distributed view of value-based decisions and involving multiple neural structures and their interaction. However, it seemed to me that this point could be expanded to (a) discuss which other structures may be involved and (b) what might be the specific role of vmPFC vs. other structures. In particular, the authors suggest that individual may use episodic memory of previous choices; is the suggestion that there may be involvement of interaction between the vmPFC and the hippocampus, in this regard? This idea would seem to fit well with recent experimental findings and theories (E.g. a recent study that examined value-based choices and DDM fits in patients with damage to the hippocampus; Bakkour, Palombo, et al., 2019. The hippocampus supports deliberation during value-based decisions. *elife.*, 8.)

Thank you for your kind comments. We wholeheartedly agree with the reviewer that the Bakkour, Palombo et al. 2019 paper is quite relevant, as is Enkavi et al. 2017, as both show that hippocampal damage leads to increases in variability in value-based choice similar to what we observed after VMF damage. As the reviewer suggests, we have added a paragraph to the discussion about what other structures may interact with the VMF to support value-based decisions, focusing on the hippocampus and striatum:

Inserted text (pg 17, lines 335 to 345) reproduced below:

On this distributed view, there are several structures that may interact with the VMF to support value-based choice. Like the VMF, neural activity in the striatum reliably scales with subjective value across dozens of functional neuroimaging studies (2). Whether striatal damage affects the variability of value-based decisions has not been studied to our knowledge, though striatal damage does impair value-based learning (35). Similar to VMF damage, damage to the hippocampus also results in more variable or inconsistent decisions (36, 37). The hippocampus has been proposed to support deliberation about value, perhaps by retrieving evidence about value from memory (36, 38). The VMF is anatomically connected to, and functionally interacts with, both striatum and hippocampus, and so either or both networks may contribute to value-based choice (39, 40). Future studies could use functional brain imaging to more directly test hypotheses about brain networks that may compensate in individuals with VMF damage.

REVIEWER COMMENTS

Reviewer #1 (Remarks to the Author):

I am satisfied with the revision, the paper is suitable and ready for publication.

Reviewer #2 (Remarks to the Author):

The authors adequately answered all my concerns.

I have no further comments.

This is an interesting paper that is suitable for publication in Nature Communications

Response to reviewers

Reviewer #1 (Remarks to the Author):

I am satisfied with the revision, the paper is suitable and ready for publication.

Reviewer #2 (Remarks to the Author):

The authors adequately answered all my concerns.

I have no further comments.

This is an interesting paper that is suitable for publication in Nature Communications

We sincerely thank all reviewers for their comments and help in improving the paper.